# The Role of Osteopontin in Atherosclerosis and Its Clinical Manifestations (Atherosclerotic Cardiovascular Diseases)—A Narrative Review

**DOI:** 10.3390/biomedicines11123178

**Published:** 2023-11-29

**Authors:** Nikolaos P. E. Kadoglou, Elina Khattab, Nikolaos Velidakis, Evangelia Gkougkoudi

**Affiliations:** Medical School, University of Cyprus, 215/6 Old Road Lefkosis-Lemesou, Aglatzia, Nicosia CY 2029, Cyprus; khattab_elina@outlook.com (E.K.); velidakis.nik@gmail.com (N.V.); gkougkoudi.evangelia@ucy.ac.cy (E.G.)

**Keywords:** osteopontin, vascular calcification, atherosclerosis, coronary artery disease, peripheral artery disease

## Abstract

Atherosclerotic cardiovascular diseases (ASCVDs) are the most common and severe public health problem nowadays. Osteopontin (OPN) is a multifunctional glycoprotein highly expressed at atherosclerotic plaque, which has emerged as a potential biomarker of ASCVDs. OPN may act as an inflammatory mediator and/or a vascular calcification (VC) mediator, contributing to atherosclerosis progression and eventual plaque destabilization. In this article, we discuss the complex role of OPN in ASCVD pathophysiology, since many in vitro and in vivo experimental data indicate that OPN contributes to macrophage activation and differentiation, monocyte infiltration, vascular smooth muscle cell (VSMC) migration and proliferation and lipid core formation within atherosclerotic plaques. Most but not all studies reported that OPN may inhibit atherosclerotic plaque calcification, making it “vulnerable”. Regarding clinical evidence, serum OPN levels may become a biomarker of coronary artery disease (CAD) presence and severity. Significantly higher OPN levels have been found in patients with acute coronary syndromes than those with stable CAD. In limited studies of patients with peripheral artery disease, circulating OPN concentrations may be predictive of future major adverse cardiovascular events. Overall, the current literature search suggests the contribution of OPN to atherosclerosis development and progression, but more robust evidence is required.

## 1. Introduction

Atherosclerotic cardiovascular diseases (ASCVDs) are the most common chronic vascular disease and a major public health problem worldwide [1]. They are characterized by inflammatory cell infiltration, like macrophage recruitment, endothelial dysfunction, lipid deposits and vascular smooth muscle cell (VSMC) migration and proliferation. Osteopontin (OPN), a multifunctional glycoprotein, has been found to be highly expressed within atherosclerotic plaques, especially at sites of macrophages and foam cell accumulation [2]. Past evidence suggests its important role in the development and progression of atherosclerosis and its potential detrimental effect on vascular homeostasis, leading to vascular injury [2,3]. OPN-induced vascular inflammation affects remodeling and precipitates the atherosclerotic process from the beginning to the eventual plaque destabilization [4]. The key role of inflammation in every stage of atherosclerosis progression is unambiguous and has been long demonstrated.

On the other hand, vascular calcification (VC) is a complex pathologic response to metabolic substances and inflammatory cells, and its incidence increases with age [5]. It has been long considered a predictor of cardiovascular morbidity and mortality. From a pathophysiological point of view, it is characterized by chronic inflammation accompanied with the deposition of minerals within the arterial wall [3]. Mineralization mechanisms and the gene expression of OPN are common in osteoblasts and chondroblasts, which can both differentiate and contribute to VC [6]. Mineralization also involves the differentiation of macrophages and VSMCs into osteoclast-like cells, which is associated with endogenous calcification inhibitors, including OPN [5]. VC is also regulated by several bone formation regulators and bone structural proteins, such as OPN, morphogenetic protein-2 (BMP-2), matrix carboxyglutamic acid protein (MGP) and osteoprotegerin (OPG), which have been found within atherosclerotic plaques [6]. VC parallels the development of atherosclerotic lesions, but on the other hand, calcium deposition on the atherosclerotic lesions makes them less prone to rupture, outlining the dual interplay between atherosclerosis and VC: atherosclerosis development and plaque vulnerability [7]. Hence, OPN, as a calcification inhibitor, suppresses calcium deposition in the VSMCs and may prevent atherosclerotic plaque calcification and reduce its stability [5]. Regarding all the actions of OPN and the complex regulation of VC, the aim of the present narrative review is to shed more light on the regulation of OPN and its precise action in the context of ASCVDs, which remain complex and, to a significant extent, unknown.

### 1.1. Literature Search

We conducted a literature search in the English language for publications in the MEDLINE and EMBASE, Web of Science, Cochrane and Google Scholar databases from 1990 to September 2023. The following search terms, for titles and abstracts, including Medical Subject Headings (MeSH), were used: osteopontin, vascular calcification, inflammation, atherosclerosis, coronary artery disease and peripheral artery disease. Three investigators (EG, NV and EK) independently performed the literature search. We included experimental studies, in vitro and in vivo, and clinical studies as well. We further limited our literature search by setting the following exclusion criteria: studies with full text unavailable, published in languages other than English, conference abstracts and interventional arms mixing of silymarin/silibinin and other substances. The reference lists of the identified articles were checked for any additional relevant articles, especially among reviews.

### 1.2. OPN: Molecular Structure and Contribution to Inflammation and Bone Homeostasis

The word osteopontin is a combination of the words “osteo”, meaning bone, and “pontin”, meaning bridge, and it is the first extracellular matrix protein identified in bone tissue [8]. OPN is a highly phosphorylated glycophosphoprotein consisting of 300–314 amino acids, including O-linked and N-linked oligosaccharides, with acidic characteristics, and it has a molecular weight of 44–75 kDa. OPN has an arginine–glycine–aspartic acid-binding zone, two heparin-binding zones, one thrombin-binding zone, one calcium-binding zone, and two terminal zones (N- and C-terminals). N-terminals bind integrin receptors, and C-terminals bind two heparin molecules and CD44 variants. Matrix metalloprotease (MMP) 3 and 7 are also bound to OPN [9]. OPN is encoded by a single gene on chromosome 4 in the human genome [10]. However, due to alternative splicing and post-translational modifications (glycosylation, serine/threonine phosphorylation, oxidation, tyrosine sulfation, calcium binding and proteolytic processing), it is expressed in different isoforms with varied molecular weights, depending on the tissue. This molecular heterogeneity explains its various functions and interactions with different proteins [11]. There are three known isoforms with different structures and functions: OPN-a (full length isoform), OPN-b (lacks exon 5) and OPN-c (lacks exon 4) [10].

OPN exists in many tissues, including saliva, bile, milk, teeth, renal tissue, brain, arteries, endothelium and skeletal muscle [9,12]. It is a multifunctional substance contributing to the secretion of interleukin-10 (IL-10), interleukin-12 (IL-12), interleukin-3 (IL-3), interferon-γ (IFN-γ), integrin avb3, nuclear factor kappa B (NF-kB), macrophage and T cells and triggering the osteoclast function. Also, OPN release is regulated by many factors, and it is enhanced by macrophages, cytokines, tumor necrosis factor-a (TNF-a) and IL-1β in pro-inflammatory conditions [9]. It is produced as a cytokine in activated macrophages and T cells, suggesting its pro-inflammatory role [13]. Peroxisome proliferator-activated receptor γ (PPARγ) is the main inhibitor of OPN gene expression [4]. Extracellular matrix proteins, including OPN, promote signal transduction into cells and contribute to their growth, transfer and their existence by adhering to them. OPN expression is induced by mechanical stress (pressure/volume overload) on bones, and it is generally not expressed in healthy cardiac tissue [14]. Stimuli known to promote OPN expression include ROS, angiotensin II, high glucose and hypoxia [15]. It seems that it is expressed by infiltrating cells a few days after a myocardial infraction, and OPN induces myocardial fibrosis and remodeling after inflammation within the myocardium [14,16]. Besides this, the expression of OPN has been detected in both macrophages and VSMCs within the atherosclerotic lesion. Evidence suggests the important role of OPN in the early stages of atherosclerosis development, including the proliferation and migration of VSMCs [17].

The pro-atherosclerotic role of OPN is not only mediated by inflammation but also through the calcification of the atherosclerotic lesions. OPN, as a protein secreted by osteoclasts, inhibits hydroxyapatite formation and promotes bone destruction. That is why it is described as a calcification inhibitor. On the other hand, OPN induces mineralization at sites of calcium accumulation. Previous researchers have described the following OPN-mediated mechanisms of mineralization: enhanced adherence and function of osseous cells, osteoblasts and osteoclasts; the accumulation of the mineralized collagen matrix; and the close binding of osteoclasts with collagen. Additional evidence suggests that OPN suppresses ectopic calcification and over-mineralization [9]. Moreover, the bone destruction induced by OPN may release calcium products to enter the bloodstream, leading to VC. Therefore, it is still unclear how OPN affects VC or under which circumstances it promotes or inhibits VC.

### 1.3. OPN: A Link between Osteoporosis and Atherosclerosis

#### 1.3.1. Atherogenetic Action of OPN in Animal Models

OPN is a multifunctional protein contributing to ASCVDs. However, its action depends on the stage of ASCVD manifestation. In an acute phase of ASCVDs, an increase in OPN may promote neovascularization and less calcification, while the persistence of OPN concentrations in a chronic phase correlates with atherosclerosis, neo-intimal hyperplasia and more adverse cardiovascular outcomes [18]. Atherosclerosis is characterized by exaggerated vascular inflammation in response to hypoxia, endothelial dysfunction and lipid overload [15]. OPN may play an important role in the process of atherogenic inflammation, but the pathophysiologic pathways of that interplay are mostly unclear [19]. Animal models have provided evidence of the atherogenic action of OPN. In particular, it involves high amounts of cytokine release, the induced migration of endothelial cells via the αvβ3 ligand, the activation of macrophages and their transformation into foam cells [9]. OPN overexpression in mice is related to medial thickening and neointimal formation, while OPN knockout (KO) mice showed atherosclerosis attenuation [20]. Also, OPN overexpression in an atherosclerosis model of OPN transgenic (TG) mice fed a high-fat diet leads to the rapid development of early fatty-streak and mononuclear cell-rich lesions and later to larger atherosclerotic plaques [21]. Another study of OPN TG mice showed significantly larger atherosclerotic lesions in OPN TG mice as compared to those in mice without OPN overexpression. Activated foamy macrophages and the in situ production of OPN within atherosclerotic plaques were significantly increased in the OPN TG mice [17].

#### 1.3.2. Factors Regulating the Atherogenetic Action of OPN

Several factors have been proposed as regulators of the OPN action in the atherosclerosis progression. Interaction with glucose-related mechanisms is of high importance. Glucose-dependent insulinotropic polypeptide (GIP) is the main incretin hormone secreted after a meal, inducing insulin secretion. It can promote OPN expression in pancreatic β-cells. A single nucleotide polymorphism (rs1800437) in the GIPR gene has been associated with increased cardiovascular risk [9]. It seems that a GIP-induced increased expression of OPN may be linked with increased atherosclerotic risk. A study of cultured rat aortic VSMCs showed that high glucose concentrations enhanced OPN secretion, leading to decreased expression of IL-10, an anti-inflammatory cytokine [22]. There are some additional pathways of hyperglycaemia-induced OPN expression, including the Rho/Rho kinase pathway in VSMCs and the nuclear factor of activated T-cells [23,24]. Besides this, an in vitro exposure to cigarette smoke promotes OPN expression in human endothelial cells [25].

Increased OPN expression in atherosclerotic plaques is related to the formation, ulceration, inflammation and instability of those plaques [4,26,27]. Its release further precipitates inflammation by attracting macrophages and T cells and their proliferation. In vitro studies showed increased expression during monocyte/macrophage differentiation [9,28]. In the context of inflammation, it regulates the recruitment of inflammatory cells and their adhesion and migration via an arginine–glycine–aspartate (RGD) sequence. OPN receptors facilitate its action, and the most known of them are integrins (avb3, avb1, avb5 and a4b1) or the splice variant of CD44 v3-v6, AT1 or AT2 [2]. OPN also promotes CD40 ligand and IFNγ expression on T cells, leading to IL-12 production and CD3 stimulation, suggesting its role in the early T helper-1 cell inflammatory response [29].

It is well known that the impairment of endothelial function reflects the early stage of atherosclerosis, where OPN could be harmful. OPN is related to decreased activity of NO synthase (NOS) within atherosclerotic lesions, contributing to endothelial dysfunction in patients with coronary artery disease (CAD) [30]. Another study demonstrated that an OPN deficiency attenuates angiotensin II-induced aortic aneurysm and atherosclerosis in animal models [31]. The coronary revascularization and angiotensin II receptor inhibitors decrease OPN plasma levels, which, in line with in vitro studies, suggests the impact of angiotensin II on enhanced OPN expression [15].

#### 1.3.3. Limited Data of the Atheroprotective Role of OPN

Limited data indicate the opposite, atheroprotective action of OPN by yielding anti-inflammatory effects. A study of cultured macrophages from hypertensive subjects showed that exogenous OPN promoted the differentiation of peripheral monocytes into an alternative anti-inflammatory phenotype and inhibited macrophage-to-osteoclast differentiation [3]. Also, they showed decreased macrophage infiltration, demonstrating OPN’s role in reducing inflammatory factor expression and attenuating osteoclast formation. In the same study, a high accumulation of osteoclasts was observed in the calcified vessels of hypertensive subjects. Therefore, in vitro limited, scarce data indicate, surprisingly, the anti-inflammatory action of OPN.

#### 1.3.4. The Controversial Role of OPN in Vascular Calcification

In the early stages of ASCVD, the atherosclerosis deposition of calcium acts as a stimulus for the further progression of atherosclerotic lesions, associated with adverse clinical outcomes [7]. On the other hand, the calcification of established atherosclerotic plaques may render them more stable, with the opposite effect on major adverse cardiovascular events (MACEs). Recent evidence showed that OPN-producing macrophages may have a protective role in VC [32,33]. OPN seems to induce the differentiation of monocytes into activated macrophages and inhibits their differentiation into osteoclasts, suppressing VC in patients [18]. Other studies suggest that OPN inhibits calcification due to hydroxyapatite crystal formation blockage and the regulation of acidification [13,18,19,34]. A direct OPN-induced inhibition of calcification in cultured bovine aortic SMCs and aortic valve calcification in vivo may be atheroprotective mechanisms [15]. OPN knockout mice on a high-phosphate diet showed that OPN prevented phosphate-induced VC [35]. An experimental study showed a significant ectopic calcification of glutaraldehyde-fixed bovine pericardium tissue in OPN−/− mice [36]. In vitro studies of VSMC calcification suggest that OPN promotes decalcification via the OPG/RANKL/RANK (receptor activator of nuclear factor kB) pathway [37]. The imbalance of this pathway is related to osteoporosis and VC [13]. All those controversial results implicate the complex mechanisms of atherosclerosis calcification, but most of them are still unknown, and those data are predominantly derived from experimental and in vitro studies, rendering it difficult to clarify the precise interaction between OPN and VC.

Finally, a loss of differentiation markers is characterized by uncontrolled proliferative activity. OPN expression parallels the induction of VSMC proliferation during the early stages of atherosclerosis, while OPN antibodies can inhibit this response. An experimental study of primary cultures of aortic mouse VSMCs showed that OPN downregulates smooth muscle actin and calpontin expression through an extracellular signaling pathway, indicating the regulating role of OPN in VSMCs’ differentiation. Also, it plays an important role in VSMC adhesion and migration [38]. Other in vitro studies have also demonstrated that OPN induces the proliferation of cultured rat VSMCs and human coronary artery smooth muscle cells [39,40]. Figure 1 and Table 1 and Table 2 summarize the pathophysiological mechanisms of OPN in atherosclerosis development.

## 2. Clinical Trials: The Relationship between OPN and ASCVDs

A plethora of clinical trials have investigated the relationship of OPN with the presence, severity and prognosis of ASCVDs like CAD and PAD. Although several OPN inhibitors have been tested in cancer and chronic kidney disease, there are no clinical data assessing OPN modifiers (inhibitors or activators) in ASCVDs [41,42,43]. On the other hand, the existing data implicate that specific medications already prescribed in patients with ASCVDs, like statins, may influence the blood levels of OPN. Therefore, the pharmaceutical modification of OPN could be useful to unravel its role in ASCVD progression. However, there is inadequate evidence in this field.

### 2.1. OPN in Coronary Artery Disease

A plethora of studies have examined the possible association of OPN levels as a biomarker of the presence, severity and prognosis of CAD and coronary calcification.

### 2.2. OPN and Presence of Coronary Artery Disease

Based on coronary angiography findings, previous cross-sectional studies have documented significantly higher levels of OPN in patients with chronic CAD than those without CAD [44,45]. However, those studies should be evaluated with caution because of the high fluctuations in the measured concentrations between studies and the confounding effect of the co-existence of aortic atherosclerosis [46]. Besides this, even higher OPN levels have been reported in patients with acute coronary syndrome (ACS) in comparison to patients with stable angina and healthy individuals [16,47,48]. In a prospective cohort study of 80 individuals with ST elevation myocardial infarction (STEMI), OPN levels were higher on admission than a group of 60 controls [49]. Notably, OPN levels tended to decrease in individuals after undergoing coronary artery graft bypass (CABG) surgery, indicating the close association between OPN and coronary patency [50]. In patients with established CAD, the relationship between OPN and CAD presence is inconsistent, and co-existing cardiovascular risk factors may be significant confounders, limiting its clinical impact [51,52].

### 2.3. OPN and Severity of Coronary Artery Disease

A limited number of published studies have investigated the association between OPN levels and the severity of CAD. The latter is quantified using mostly the number of vessels with stenosis greater than 50%. Two studies from different research groups on individuals undergoing elective coronary angiography showed a positive correlation between OPN levels and the number of affected coronary vessels [44,46]. In particular, there was an almost 50% increase in OPN levels from patients with one-vessel disease to those with three-vessel disease (from 540 ± 293 ng/mL to 758 ± 416 ng/mL) [44]. Lastly, the log-transformed OPN levels were independently related to the Gensini score, an index of CAD severity, in patients with known CAD undergoing coronary angiography [53]. However, in patients admitted with ACS, OPN was independent of the number of affected vessels [47]. Perhaps the acute destabilization of coronary plaque and the OPN release superimpose the severity of the underlying CAD.

### 2.4. OPN and Coronary Calcification

Regarding OPN as a VC inhibitor, many researchers have hypothesized a potential inverse relationship between OPN and the calcification of atherosclerotic coronary plaques. Increased OPN levels have been found among patients with CAD and calcified plaques, but that relationship was lost in a multivariate analysis [44]. The introduction of the CAC score in clinical practice has provided a more accurate, quantifiable measure of atherosclerotic plaque calcification. Among diabetic and non-diabetic patients undergoing computed tomography (CT), high OPN levels, either as a continuous variable or in the top quartile, are independently associated with the Agatston score [54,55]. Individuals with a CAC score > 0 had higher OPN levels than their counterparts with CAC = 0, and importantly, the OPN levels paralleled the Agatston score elevation [56,57]. A large prospective cohort study with 1207 participants with known CAD and a 4-year follow-up period identified higher OPN levels at baseline in participants with a CAC score of 0–100 or a CAC score > 100 in comparison to participants with a CAC score of 0 [58]. Nevertheless, the relationship between OPN and the CAC score is weak, and during the ROC analysis, the cut-off value of 18.45 ng/mL provided a sensitivity of 72% and a specificity of 73% for the prediction of calcification. An ROC analysis in another study proposed a cut-off value of 48.5 ng/mL of OPN among individuals with type 2 diabetes mellitus for the prediction of calcification with a sensitivity of 79.8% and a specificity of 75.2% [59]. Perhaps several medications lessen the power of their association. In a cross-sectional study of asymptomatic individuals with an intermediate risk for coronary events, OPN was found to be associated with the CAC score after a multiple logistic regression (OR: 1.63; CI: 1.10–2.40, *p* = 0.014) only in the subgroup not receiving statins and/or renin-angiotensin system inhibitors [60]. Therefore, the administration of the aforementioned drugs may alter the expression of OPN and attenuate its link with the CAC score [61,62].

### 2.5. OPN and Prognosis of Coronary Artery Disease

A recently published large cohort study with 666 participants undergoing a PCI after an anterior myocardial infarction and who were followed up for at least 6 years documented higher OPN levels among patients with re-hospitalization. Particularly, levels above 4.81 ng/mL were associated with a 3.2 times increased risk of re-hospitalization in comparison to levels below 2.60 ng/mL (HR: 3.2; CI: 1.23–8.33, *p* = 0.017) after multivariate adjustments [63]. A large study of 730 patients presented with STEMI also identified a positive association with higher OPN levels and cardiovascular mortality (HR: 1.08; CI: 1.04–1.11) and all-cause mortality (HR: 1.09; CI: 1.04–1.11) for an increase of 10 ng/mL, while no statistically significant association was found for heart failure or new myocardial infarction [64]. On the contrary, in this prospective cohort, the OPN levels on admission and the peak values did not seem to have an association with the occurrence of MACEs or left ventricular dysfunction during the hospitalization period [49]. Among patients with ACS, higher OPN levels were associated with the rapid progression of coronary atheromatic plaques when left untreated or with in-stent re-stenosis after 6 months when a PCI was performed [16]. Besides this, patients with stable CAD and high OPN levels had a greater risk of adverse cardiovascular events during the mean follow-up period of 2.7 years (HR: 1.88; CI: 1.35–2.62, *p* < 0.001) [65]. In 101 individuals with CAD, when the researchers compared high versus low OPN levels (a cut-off value of 55 ng/mL), they noticed a 2.88 times higher risk of MACE after 3 years of follow-up (HR: 2.88; CI: 1.09–7.58, *p* = 0.032) [66]. In a small prospective cohort study of 130 individuals undergoing a PCI, those with OPN levels above 500 ng/mL had lower event-free survival in comparison to those with OPN blood concentrations below 500 ng/mL. Moreover, an increase in OPN levels by 100 ng/mL was associated with a 30% greater risk of MACEs during the follow-up (HR: 1.3; CI: 1.1–1.5, *p* < 0.005) [67]. The same research team earlier revealed 70% higher odds of restenosis for every 100 ng/mL increase in OPN levels in individuals undergoing elective coronary angiography (OR: 1.7; CI: 1.2–2.5) [68].

Recently, McCarthy et al. tried to create a score for individuals undergoing coronary angiography for the prediction of MACEs [69]. OPN was included in the final model, among other biomarkers, strengthening the indications that OPN may have a predictive role for CAD. It is evident that OPN may be an indicator of rapid disease progression and a more active atherosclerotic process.

The major disadvantage of most clinical studies is the lack of long-term follow-up of the participants and the use of heterogenous populations across the different studies, which makes more difficult to draw firm conclusions. Furthermore, the relatively small number of participants in each study may attenuate the statistical power of their results. However, OPN seems to be a promising biomarker in terms of the presence and severity of CAD as well as its prognosis, while conflicting data exist for intra-plaque calcification. It is necessary to carry out larger studies to validate the aforementioned results because the reported cut-off values varied widely among studies. Table 3 shows the role of OPN in ASCVDs with regards to coronary arteries (Table 3).

## 3. OPN in Peripheral Artery Disease (PAD)

### 3.1. OPN and Carotid Atherosclerosis

A few studies have investigated the possible association between OPN levels and the presence of carotid atherosclerosis. Those studies indicated a positive relationship between higher OPN levels and the presence of carotid atherosclerosis [71,72]. Regarding preclinical carotid atherosclerosis, carotid intima-media thickness (CIMT), individuals with psoriasis or various stages of chronic kidney disease, hypertension or diabetes mellitus appeared with a higher CIMT along with OPN levels [73,74,75,76]. In contrast, other studies have failed to identify a significant association between OPN levels and preclinical carotid atherosclerosis [77,78,79]. This is the case in a study with systemic lupus erythematosus (SLE) patients, but the lack of a relationship was attributed, by the authors, to the young age of the participants, the SLE duration and a possible confounding effect of warfarin use [77].

In the case of patients with established carotid atherosclerotic plaques, they showed higher serum OPN levels [67,75]. OPN seemed to be inversely correlated with the grey-scale median (GSM), an ultrasound index of carotid plaque echogenicity and thereby stability [71]. Interestingly, the examination of carotid plaque specimens after endarterectomy indicated higher intra-plaque concentrations of OPN in patients with recent neurological symptoms than their asymptomatic counterparts [36]. In another study, a histological analysis of endarterectomy samples of patients with ischemic stroke revealed a positive correlation between the intra-plaque content of pro-inflammatory markers, such as the macrophage CD86, and serum OPN levels. Besides this, OPN levels predicted the incidence of MACEs for a 24-month follow-up period after carotid endarterectomy. Notably, OPN levels above 70 ng/mL were linked with a 12.5 times higher risk of ischemic stroke in those patients (HR: 12.52; CI: 1.40–111.93) [68]. Using an ROC analysis, a cut-off value of 50 ng/mL of OPN stressed the detection of carotid atherosclerosis with adequate sensitivity (70%) and specificity (69%) among patients with end-stage kidney disease. Higher cut-off values may increase the specificity (81%) but not the sensitivity (66%) [75]. Clinical studies in patients with established carotid atherosclerosis requiring [80] or not requiring [81,82] revascularization have reported that aggressive lipid lowering with statins suppresses carotid plaque vulnerability in parallel with a favorable decline in serum OPN levels. Regarding statins as the cornerstone of the pharmaceutical therapy of ASCVDs, those findings outline the “pleiotropic” actions of statins, targeting another mediator of atherosclerotic plaque destabilization. Although those results indicate the possible role of OPN in the determination of carotid atherosclerosis presence and vulnerability, they should be considered with caution due to the relatively small number of participants and the high variance of the population characteristics. More and larger studies are necessary to draw safer conclusions.

### 3.2. OPN and Lower Extremity Artery Disease

To our knowledge, a limited number of studies have evaluated the correlation between OPN levels and the presence of PAD in the lower extremities [83]. Like carotids, the majority of those studies have identified a possible association between higher OPN levels and established PAD. Recently, an increase in OPN by 10 ng/mL was associated with a 16% increase in the risk of PAD. A negative correlation between OPN levels and the ankle–branchial index (ABI) further supported a possible link between OPN levels and PAD severity in diabetic and non-diabetic patients [83,84]. Regarding the prognostic power of OPN, a cut-off value of 126 ng/mL for OPN predicted mortality among individuals with PAD undergoing percutaneous revascularization. That cut-off value showed a sensitivity of 80% and a specificity of 70% [69]. In a prospective cohort of 203 patients with PAD undergoing transcutaneous revascularization, the baseline values of OPN were significantly higher than in 78 controls. It is important to mention that in the multivariate logistic regression, the association between PAD presence and OPN at baseline did not reach statistical significance throughout the whole cohort of PAD patients and controls [85]. However, during the 12-month follow-up, high OPN levels were detected in patients with MACEs. Interestingly, a large cohort of patients with abdominal aortic aneurysms showed significantly higher levels of OPN than healthy controls before undergoing endovascular repair [86].

Table 4 shows the role of OPN in ASCVDs with regards to periphery arteries (Table 4).

## 4. Discussion and Conclusions

OPN is a multifunctional glycoprotein expressed by a variety of cell types, including cardiac cells, and plays an important role in several biological processes, like atheromatosis and VC. In the present narrative review, most of the studies demonstrated that OPN exerts a pro-inflammatory role in ASCVDs and shows a complex interplay with VC. The underlying mechanisms require further investigation, and the studies’ findings require further validation. In clinical terms, most of the studies supported the notion of OPN as a biomarker of the presence, severity and prognosis of CAD. Notably, OPN levels were further increased in the acute phase of ACS. In a limited number of cohorts with carotid stenosis, serum OPN levels have been considered a biomarker of prognosis and plaque stability, while in patients with PAD of the lower extremities, serum OPN is associated with PAD presence and the MACE incidence after revascularization.

Further investigation is needed to determine the discrete role of the multiple OPN isoforms’ expression in several vascular diseases.

## Figures and Tables

**Figure 1 biomedicines-11-03178-f001:**
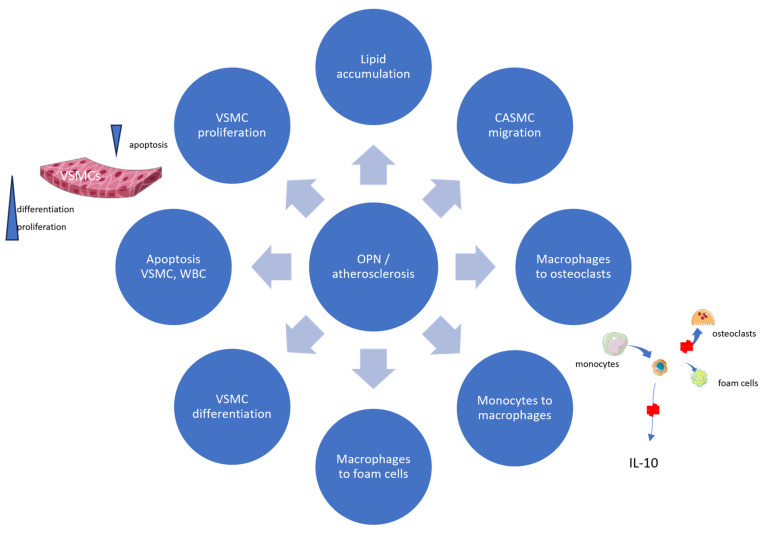
Pathophysiology mechanism of OPN in atherosclerosis development. CASMCs, coronary artery smooth muscle cells; IL-10, interleukin 10; OPN, osteopontin; VSMCs, vascular smooth muscle cells; WBC, white Blood Cells.

**Table 1 biomedicines-11-03178-t001:** The pathophysiologic mechanisms of OPN contribution to atherosclerosis based on in vitro studies.

Cell Types/Origin	OPN Assessment or OPN Regulation (+Duration)	Key Effects/Mechanisms	Reference
RAW 264.7 macrophage-like cells	Cells w/silenced OPN expression using RNAi (siRNA sequence: GTTTCACAGCCACAAGGA, MGW RNAi selection tool)	OPN deficiency caused by - Altered macrophage phenotype - ↑ macrophage apoptosis - ↓ IL-12 production	Nyström T (2007) [28]
Primary cultured macrophages from 70 hypertensive pts w/VC based on abdominal CT scan	Cells were induced by OPN (0.5 μmol/L), and then total RNA was extracted. Duration: 24 h	- Regulation of monocyte/macrophage phenotypic differentiation - ↓ inflammatory factors expression ↓ macrophage-to-osteoclast differentiation - Macrophage recruitment	Ge Q, Ruan CC (2017) [3]
CEA samples; 24 pts w/recent (≤6 wks) stroke vs. 22 asymptomatic pts with significant carotid stenosis	Carotid plaque specimens cultured for 8 days Incubation w/ibesartan (1 mg/mL)	↑ OPN in symptomatic and unstable atherosclerotic plaques ↓ OPG by in vitro ibesartan blockade	Golledge J (2004) [37]
Autopsy human cases (*n* = 25): foam-cells from aortic atherosclerotic lesions	Assessment of OPN expression	↑ OPN expression depending on the atherosclerosis stage	Ikeda T (1993) [26]
Primary cultures of aortic mouse VSMCs (C57BL/6 mice)	OPN up-regulation model using the pIREShrGFP-1a-OPN mammalian expression plasmid. Duration: 72 h	↓ Reduction in a-SM actin and calponin ↑ VSMCs’ differentiation and proliferation	Gao H (2012) [38]
Both cultured human CASMCs and atherectomy specimens from coronary arteries of 13 pts undergoing DCA and samples of normal coronary arteries (autopsy) from 6 pts	OPN mRNA expression from DCA tissues and cultured CASMCs OPN plasma levels obtained 24 h and 4 wks after DCA	Migration of CASMCs to the site of angioplasty injury, with subsequent proliferation causing restenosis	Panda D (1997) [39]
SMCs cultures obtained from normal rat aorta	OPN mRNA assessed. Duration: 3 days	↑ Arterial SMCs proliferation associated with atherosclerosis and restenosis	Gadeau AP (1993) [40]
SMCs from aortas and collateral branches of 12 wk-old SDN TG mice ApoE−/− vs. non TG mice Aortic aneurysms and non-aneurysmal aortic specimens from pts undergoing elective surgery	SMCs were cultured in a 24-well plate overnight then treated with PBS or Ang II (1 μM) for 48 h. Then, the cells were infected with an adenovirus overexpressing human Nox4 or an empty adenovirus (control)	Ang II increased the expression of Nox4, OPN ↑ SMC proliferation, migration and macrophage adhesion to SMCs promoted by OPN ↑ Nox4 and OPN expressions in human aortic aneurysms	Yu W (2020) [31]

Ang II, angiotensin II; ApoE−/−, ApoE deficiency; a-SM actin, alpha smooth muscle actin; CEA, carotid endarectomy; CASMCs, coronary artery smooth muscle cells; DCA, directional coronary atherectomy; h, hours; IL-12, interleukin 12; Nox4DN, human dominant negative form of Nox4; OPN, osteopontin; OPG, osteoprotegerin; pts, patients; RNAi, ribonucleic acid interference; siRNA, small interfering RNA; SDN, smooth muscle-specific Nox4 dominant negative; SMCs, smooth muscle cells; TG, transgenic; VC, vascular calcification; VSMCs, vascular smooth muscle cells; w/, with; wks, weeks.

**Table 2 biomedicines-11-03178-t002:** The pathophysiologic mechanisms of OPN contribution to atherosclerosis based on in vivo studies.

Animals (Gender, Age, Models)	Type of Intervention and Duration	Key Effects/Mechanisms	Reference
OPN TG mice on atherogenic or standard diet versus wild type C57BL/6 mice	1st group: atherogenic diet from 6 to 18 wks (*n* = 8) vs. control (*n* = 21) 2nd group: standard diet from 6 to 18 wks (*n* = 6) vs. control (*n* = 6) 3rd group: standard diet from 6 to 36 wks (*n* = 14) vs. controls (*n* = 22) Duration: 3 months Evaluation of atherosclerotic lesions at 18 or 36 wks	↓ atherosclerosis in OPN TG mice ↑ transformation of macrophages→foam cells Macrophage recruitment	Chiba S (2002) [17]
C57BL/6 LDLR−/− mice or FVB/N ApoE−/− mice	Male mice (10 wks old) randomized to receive Ang II (1.44 mg/kg/day) or saline via sc osmotic minipumps for 4 wks, andthen aneurysm specimens were obtained	↓ aneurysm and atherosclerosis development ↑ transformation macrophages→foam cells	Yu W (2020) [31]
Heterozygous OPN TG and non TG mice	High-fat/cholesterol diet for 16 wks Evaluation of atherosclerotic lesions at the end	↓ atherosclerosis (early fatty streak lesions promoted by OPN) Macrophage recruitment	Isoda K (2003) [21]
Female OPN KO/DBA/2J mice vs. wild type mice (WT)	High phosphate (HP) diet vs. normal phosphate (NP) diet for 11 wks Groups: OPN KO + HP (*n* = 10) OPN KO + NP (*n* = 10) WT + HP (*n* = 10) WT + NP (*n* = 10)	↓ VC ↑ plaque vulnerability	Paloian NJ (2005) [35]
Female OPN−/− mice Bovine GFBP tissue subcutaneously implanted Full-length recombinant rat histidine-fused OPN (rat His-OPN) was generated in bacteria	100 μL of an OPN solution (850 lg/mL or 21.25 lM OPN in sterile PBS) or PBS alone was injected every 2 days in 8 OPN−/− mice Duration: 7 days Levels of calcification were assessed at 3, 7, 14 and 30 days Afterwards, soluble injection of exogenous rat His-OPN	↓ ectopic calcification ↓ VC ↑ plaque vulnerability Direct rescue of calcification phenotype by (rat His-OPN)	Ohri R (2005) [36]
Female mice OPN+/+ apoE−/− (*n* = 27), OPN+/− apoE−/− (*n* = 29) OPN−/− apoE−/− (*n* = 37)	Normal chow diet Duration: 36 wks	↓ VC ↑ plaque vulnerability	Matsui Y (2003) [20]

Ang II, angiotensin II; ApoE−/−, ApoE deficiency; GFBP, glutaraldehyde-fixed bovine pericardium; KO, knockout; LDLR−/−, low-density lipoprotein receptor deficiency; Nox4DN, human dominant negative form of Nox4; OPN, osteopontin; SMCs, smooth muscle cells; SDN, smooth muscle-specific Nox4 dominant negative; sc, subcutaneous; TG, transgenic; VC, vascular calcification; wks, weeks; w/, with; w/o, without.

**Table 3 biomedicines-11-03178-t003:** The relationship between OPN and CAD based on clinical studies.

Association	Clinical Cohort	Results	Reference
OPN and presence of CAD	Elective coronary angiography (*n* = 178). Groups: CAD (67 ± 8 yrs) vs. no CAD (62 ± 8 yrs)	↑ OPN levels associated with CAD presence (OR: 1.21; 95% CI: 1.05, 1.39)	Ohmori (2003) [44]
	Complaint of ischemic chest pain (*n* = 120). Groups: CAD (55 ± 6.5 yrs) vs. no CAD (52.7 ± 7.8 yrs)	↑ OPN levels associated with CAD presence	Abdel-Azeez (2010) [45]
	Coronary angiography for suspected or known CAD (*n* = 136, 64 ± 9 yrs)	↑ OPN levels associated with CAD presence	Momiyama (2010) [46]
	ACS, stable angina or control (*n* = 108). Groups: ACS (57 ± 11 yrs), stable angina (51 ± 13), control (55 ± 5 yrs)	↑ OPN levels in ACS group than stable angina and control	Coskun (2006) [47]
	ACS or CCS (*n* = 77, 61 ± 10 yrs)	↑ OPN levels in ACS than CCS pts	Mazzone (2011) [16]
	Acute STEMI (*n* = 140). Groups: STEMI (55.5 ± 11 yrs) vs. control (53.3 ± 10.9 yrs)	↑ OPN levels on admission in STEMI pts ↑ OPN levels associated with higher incidence of STEMI	Okyay (2011) [49]
	Elective CABG (*n* = 50, 63 ± 10 yrs)	↓ OPN levels after CABG	Sbarouni (2012) [50]
	Asymptomatic (*n* = 544) with low risk factors (LRF) or multiple risk factors (MRF). Groups: (1) LRF + CAD (63.8 ± 7.2 yrs). (2) LRF + no CAD (57.5 ± 8.5 yrs). (3) MRF + CAD (58.2 ± 8 yrs). (4) MRF + no CAD (62.5 ± 7.1 yrs)	↑ OPN levels in LRF + CAD group (OR: 8.42; 95% CI: 1.51, 46.83) ↔ OPN levels in MRF + CAD group	Carbone (2022) [51]
	Ischemic heart disease (*n* = 82). Groups: stable angina (63.6 ± 9.2 yrs) vs. ischemia on stress test (60.4 ± 7.3 yrs)	↔ OPN levels regardless of history of MI or CAD	Göçer (2020) [52]
OPN and severity of CAD	Elective coronary angiography (*n* = 178). Groups: CAD (67 ± 8 yrs) vs. no CAD (62 ± 8 yrs)	↑ OPN levels associated with the number of vessels with stenosis > 50% (r = 0.35, *p* < 0.05) or >25% (r = 0.43, *p* < 0.05)	Ohmori (2003) [44]
	Coronary angiography for suspected or known CAD (*n* = 136, 64 ± 9 yrs)	↑ OPN levels associated with the number of affected vessels (r = 0.23, *p* < 0.05)	Momiyama (2010) [46]
	ACS or stable angina or control (*n* = 108). Groups: ACS (57 ± 11 yrs), stable angina (51 ± 13 yrs), control (55 ± 5 yrs)	No association between OPN levels and either the number of affected vessels or the extent of coronary stenosis	Coskun (2006) [47]
	Suspected CAD (*n* = 409). Groups across the number of affected arteries (0, 1, 2, 3)	No association between OPN levels and the Gensini score (OR: 1.03; 95% CI: 0.97, 1.08)	Tousoulis (2012) [53]
OPN and coronary calcification	T2DM (62.1 ± 9.3) vs. non-diabetic pts (59.6 ± 6.3 yrs)	↑ OPN levels associated with ↑ Agatston score (r = 0.251, *p* < 0.05)	Ishiyama (2009) [54]
	T2DM and CAD (*n* = 126). Groups: T2DM + CAD (59.1 ± 2.8 yrs) vs. non-T2DM pts with CAD (56.6 ± 5.4 yrs)	↑ OPN levels associated with ↑ Agatston score (OR: 8.34; 95% CI: 3.22, 11.40 (4th vs. 1st OPN quartile, for non-T2DM pts), OR: 8.34; 95% CI: 3.14, 14.20 (4th vs. 1st OPN quartile, for T2DM pts))	Berezin (2013(b)) [59]
	Asymptomatic CAD (*n* = 126, 58.3 ± 9.6 yrs)	↑ OPN levels associated with ↑ Agatston score (OR: 1.14; 95% CI: 1.12, 1.25)	Berezin (2013(a)) [55]
	CAD (*n* = 112, 59.8 (55, 70) yrs) *	↑ OPN levels in mild Agatston score in comparison to severe Agatston score	Golovkin (2016) [57]
	Suspected CAD (*n* = 64, 45.5 ± 10.9 yrs)	↑ OPN levels associated with ↑ CAC score (r = 0.35, *p* < 0.05)	Uz (2009) [56]
	Suspected CAD (*n* = 178) Groups: CAD (67 ± 8 yrs) vs. no CAD (62 ± 8 yrs)	↑ OPN levels associated with number of calcified vessels (r = 0.26, *p* < 0.05)	Ohmori (2003) [44]
	Asymptomatic at intermediate risk for coronary events (*n* = 80, 56 ± 10 yrs)	↑ OPN levels associated with ↑ CAC score (in statin- and/or RASI-free pts) (OR: 1.63; 95% CI: 1.10, 2.40)	Aryan (2009) [60]
	Asymptomatic from general population (*n* = 316, 57 ± 7.9 yrs)	No association between OPN and CAC score (OR: 1.07 (95% CI: 0.86, 1.34) for hypertensive; OR: 0.98 (95% CI: 0.58, 1.67) for normotensive)	Stępień (2012) [62]
OPN and prognosis of CAD	AMI or CAD with previous PCI or CABG (*n* = 666, 66.3 ± 12.7 yrs)	↑ OPN levels among pts who needed re-hospitalization (HR: 3.20 (95% CI: 1.23, 8.33) between 3rd and 1st OPN tertile) No association between OPN and cardiovascular death (HR: 0.77; 95% CI: 0.43, 1.38)	Cheong (2023) [63]
	CAD with previous PCI (*n* = 91). Groups: no restenosis (65 ± 9 yrs) vs. restenosis (67 ± 7 yrs)	↑ OPN levels associated with increased death (HR: 1.30 (95% CI: 1.10, 1.50) for MACE)	Kato (2009) [67]
	STEMI (*n* = 730). Groups: OPN < 100 μg/L (61.5 ± 11.6 yrs), OPN ≥ 100 μg/L (65.1 ± 12.2 yrs)	↑ OPN levels associated with cardiovascular mortality and all-cause mortality (HR: 1.08 (95% CI: 1.04, 1.11) for CV mortality; HR: 1.09 (95% CI: 1.06, 1.11) for all-cause mortality)	Bjerre (2013) [64]
	ACS or CCS (*n* = 77, 61 ± 10 yrs).	↑ OPN levels associated with rapid progression of atheromatic plaques or in-sent re-stenosis	Mazzone (2011) [16]
	Stable angina (*n* = 799, 64.9 ± 9.6 yrs).	↑ OPN levels associated with future adverse CV events (HR: 1.88 (95% CI: 1.35, 2.62))	Minoretti (2006) [65]
	CAD (*n* = 101, 67 ± 10 yrs).	OPN > 55 ng/mL associated with adverse outcome (HR: 2.88 (95% CI: 1.09, 7.58))	Georgiadou (2010) [66]
	CAD with suspected restenosis (*n* = 150). Groups: restenosis (65 ± 10 yrs), no restenosis (61 ± 9 yrs), no PCI (63 ± 9 yrs)	↑ OPN levels associated with risk for restenosis (OR: 1.70 (95% CI: 1.20, 2.50))	Kato (2006) [68]
	CAD or PAD (*n* = 1205, 40–82 yrs **)	↑ OPN levels associated with all-cause death	Lin (2019) [69]
	Coronary ± peripheral angiography (*n* = 649). Groups: MACE (73 ± 11 yrs) vs. no MACE (65.3 ± 11.5 yrs)	↑ OPN levels associated with MACE OPN included in a scoring system for MACE prediction	McCarthy (2017) [70]
	Acute STEMI (*n* = 140). Groups: STEMI (55.5 ± 11 yrs) vs. control (53.3 ± 10.9 yrs)	No association between baseline and peak (3rd day) OPN levels with occurrence of MACE	Okyay (2011) [49]

ACS; acute coronary syndrome, AMI; anterior myocardial infarction, CABG; coronary artery bypass graft, CAC; coronary artery calcium, CAD; coronary artery disease, CCS; chronic coronary syndrome, CCTA; coronary computed tomography angiography, CV; cardiovascular, HF; heart failure, HR; hazard ratio, MACE; major adverse cardiovascular event, MI; myocardial infarction, NS; non-significant, OPN; osteopontin, OR; odds ratio, PAD; peripheral artery disease, PCI; percutaneous coronary intervention, pts; patients, RASIs; renin–angiotensin system inhibitors, SD; standard difference, STEMI; ST elevation myocardial infarction, T2DM; type 2 diabetes mellitus. yrs; years. * Age is presented as median and interquartile range (25%, 75%). ** Age is presented as a range (min, max).

**Table 4 biomedicines-11-03178-t004:** The relationship between OPN and PAD based on clinical studies.

Association	Clinical Conditions	Results	Studies
**Carotid Atherosclerosis**			
Preclinical carotid atherosclerosis	Psoriasis, CKD, hypertension, DM, SLE or general population	↑ or ↔ association between OPN levels and CIMT ↑ OPN levels associated with lower Vd/Vs	Robati (2016) [73] Chaitanya (2018) [74] Behairy (2022) [75] Kurata (2006) [76] Ishiyama (2009) [54] Wirestam (2021) [77] Wendelin (2011) [79]
Presence of carotid atherosclerosis	CAD and carotid stenosis	↑ or ↔OPN levels in pts with carotid concomitant CAD and carotid stenosis	Kadoglou (2008) [67] Del Toro (2021) [75]
Plaque composition	Ischemic stroke and carotid plaques Pts undergoing CEA	↑ OPN levels associated with intra-plaque infiltration of pro-inflammatory markers in pts with ischemic stroke ↑ OPN intra-plaque levels in symptomatic pts	Carbone (2018) [68] Golledge (2004) [37]
Calcification	Carotid stenosis > 50%	↑ OPN levels associated with ↓ GSM score	Kadoglou (2008) [71]
Prognosis of carotid atherosclerosis	Ischemic stroke and carotid plaques	↑ OPN levels associated with MACE incidence	Carbone (2018) [72]
Severity of carotid atherosclerosis	CAD and carotid stenosis	↔ Association between OPN levels and the severity of carotid stenosis	Del Toro (2021) [78]
**PAD—lower extremities**			
Presence of PAD	Established PAD	↑ OPN levels associated with PAD prevalence	Kadoglou (2022) [85] Koshikawa (2009) [83]
	DM	↑ OPN levels associated with ↑ PAD prevalence (in both diabetic and non-diabetic subgroups) ↑ OPN levels associated with ↓ ABI values	Eleftheriadou (2020) [84]
Prognosis of PAD	Percutaneous intervention for established PAD	↑ OPN levels associated with higher mortality rates ↑ OPN levels in pts with MACE	Lin (2019) [69] Kadoglou (2022) [85]

ABI; ankle–branchial index, CIMT; carotid intima-media thickness, CEA, carotid endarterectomy; CKD; chronic kidney disease, DM; diabetes mellitus, GSM; gray-scale median, MACE; major adverse cardiovascular event, SLE; systemic lupus erythematosus, Vd/Vs; relative diastolic flow of the common carotid artery.

## Data Availability

Not applicable.

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
