# Peer review of "The Role of Osteopontin in Atherosclerosis and Its Clinical Manifestations (Atherosclerotic Cardiovascular Diseases)—A Narrative Review"

_biomedicines, 2023, doi:10.3390/biomedicines11123178_

Round 1
Reviewer 1 Report
Comments and Suggestions for Authors
The review manuscript, titled “The role of osteopontin (OPN) in atherosclerosis and its clinical manifestations (atherosclerotic cardiovascular diseases -ASCVDs) – a narrative review” by Kadoglou et al., well summarizes current understanding of OPN in ASCVDs. However, some points described below should be addressed before publication to improve the manuscript.
Line 17
‘VSMCs’ should be spelled out.
Line 33
To avoid misunderstanding, ‘N-linked multifunctional glycoprotein’ should be ‘multifunctional glycoprotein.’ Because OPN could be modified with not only N-linked glycans but also O-linked glycans.
Line 83
If OPN does not covalently bind to MMPs, the term ‘linked’ should be changed to ‘bound.’
Line 94
‘avB3’ should be ‘avb3.’ Please review the entire manuscript and ensure the proper use of Greek letters. This comment applies to other parts of the manuscript, such as Line 96.
Line 111
I had difficulty understanding the term ‘a derivative of osteoclasts.’ Does it refer to ‘a protein secreted by osteoclasts’?
Line 122-
Section 1.3 appears to be somewhat lengthy. If feasible, please consider the possibility of dividing it into smaller sections for better readability.
Line 126
‘persistence’ should be removed.
Line 157
‘increased expression’ should be changed to ‘increased OPN expression.’
Lines 173-176
Please add a reference at the end of this sentence.
Line 178-
The sentence from line 173 describes the contribution of OPN to osteoclast differentiation in hypertensive patients. Does the sentence from line 178 differ from that, and are the cells described in this sentence derived from healthy individuals? Please clarify.
Lines 356-358
Please add a reference at the end of this sentence.
Author Response
Reviewer 1.
The review manuscript, titled “The role of osteopontin (OPN) in atherosclerosis and its clinical manifestations (atherosclerotic cardiovascular diseases -ASCVDs) – a narrative review” by Kadoglou et al., well summarizes current understanding of OPN in ASCVDs. However, some points described below should be addressed before publication to improve the manuscript.
We really appreciate the reviewer’s suggestion which are very helpful to improve the manuscript and we have accepted all the following suggestions and made the corrections accordingly:
-Line 17: ‘VSMCs’ should be spelled out.
-Line 33: To avoid misunderstanding, ‘N-linked multifunctional glycoprotein’ should be ‘multifunctional glycoprotein.’ Because OPN could be modified with not only N-linked glycans but also O-linked glycans.
- Line 83: If OPN does not covalently bind to MMPs, the term ‘linked’ should be changed to ‘bound.’
-Line 94: ‘avB3’ should be ‘avb3.’ Please review the entire manuscript and ensure the proper use of Greek letters. This comment applies to other parts of the manuscript, such as Line 96.
Line 111: I had difficulty understanding the term ‘a derivative of osteoclasts.’ Does it refer to ‘a protein secreted by osteoclasts’?
Line 122-:Section 1.3 appears to be somewhat lengthy. If feasible, please consider the possibility of dividing it into smaller sections for better readability.
Line 126:‘persistence’ should be removed.
Line 157: ‘increased expression’ should be changed to ‘increased OPN expression.’
Lines 173-176: Please add a reference at the end of this sentence.
The sentence from line 173 describes the contribution of OPN to osteoclast differentiation in hypertensive patients. Does the sentence from line 178 differ from that, and are the cells described in this sentence derived from healthy individuals? Please clarify.
Line 181-184 (previous 173-178):Thanks for the comment, because it is the same study with 2 important observations. We have corrected
Lines 356-358: Please add a reference at the end of this sentence.
New lines 396 we have added reference 82.
Reviewer 2 Report
Comments and Suggestions for Authors
In this review manuscript, the authors have summarized the existing literature on the role of osteopontin in atherosclerotic vascular diseases, including both experimental and clinical studies.
I recommend the following changes to the manuscript:
1. Separate the tables summarizing in vitro and in vivo models into two distinct tables. For in vitro studies, authors should provide the following information for each study: cell types used (macrophages or SMCs or any other cell types), including the exact origin of the cells; dosages of osteopontin used to stimulate the cells; approaches used to inhibit endogenous osteopontin expression (siRNAs) or small molecular inhibitors; duration of the experiments (e.g., 24h, 48h, or other time points); and key effects on the studied cells (such as proliferation, migration, or synthesis of other factors like collagens). For in vivo studies, authors should create another table containing the following details: animals used (rats or mice), their genetic background, gender, and age; interventions (gene knockout, with specific details if applicable, or osteopontin inhibiting/activating approaches, including neutralizing antibodies, recombinant osteopontin proteins, or small molecular inhibitors, with exact dosages and methods of administration); the animal model details (e.g., model type, such as AngII infusion), duration (in weeks or days), and key effects. In both tables, authors should allocate separate rows for each study.
2. In the text, authors have provided hazard ratio (HR) and odds ratio (OR) values, as well as circulating osteopontin values from the studies they cite. I suggest that these values be included in the tables dedicated to the clinical studies. Additionally, authors should furnish more details about the clinical studies they cite, including the number of subjects, age, and gender of the subjects, values of osteopontin in each group (including control groups) and specific diagnoses, along with key findings related to osteopontin levels in the study (e.g., correlations).
3. The authors should include a separate section to discuss therapeutic options for inhibiting or activating osteopontin in atherosclerotic diseases. Are there any ongoing clinical trials aimed at inhibiting or activating osteopontin in this patient group?
4. The authors should create a diagram illustrating the osteopontin signaling pathways in the pathogenesis of atherosclerotic diseases.
Author Response
Reviewer 2
In this review manuscript, the authors have summarized the existing literature on the role of osteopontin in atherosclerotic vascular diseases, including both experimental and clinical studies. I recommend the following changes to the manuscript:
- Separate the tables summarizing in vitro and in vivo models into two distinct tables. For in vitro studies, authors should provide the following information for each study: cell types used (macrophages or SMCs or any other cell types), including the exact origin of the cells; dosages of osteopontin used to stimulate the cells; approaches used to inhibit endogenous osteopontin expression (siRNAs) or small molecular inhibitors; duration of the experiments (e.g., 24h, 48h, or other time points); and key effects on the studied cells (such as proliferation, migration, or synthesis of other factors like collagens). For in vivo studies, authors should create another table containing the following details: animals used (rats or mice), their genetic background, gender, and age; interventions (gene knockout, with specific details if applicable, or osteopontin inhibiting/activating approaches, including neutralizing antibodies, recombinant osteopontin proteins, or small molecular inhibitors, with exact dosages and methods of administration); the animal model details (e.g., model type, such as AngII infusion), duration (in weeks or days), and key effects. In both tables, authors should allocate separate rows for each study.
Following reviewer’s suggestion we have separated into 2 distinct tables in vitro and in vivo data Table 1. We also allocated separate rows for each study. The columns have been created based on reviewer’s suggestion.
- In the text, authors have provided hazard ratio (HR) and odds ratio (OR) values, as well as circulating osteopontin values from the studies they cite. I suggest that these values be included in the tables dedicated to the clinical studies. Additionally, authors should furnish more details about the clinical studies they cite, including the number of subjects, age, and gender of the subjects, values of osteopontin in each group (including control groups) and specific diagnoses, along with key findings related to osteopontin levels in the study (e.g., correlations).
This is an important suggestion. We have added the HR and OR in the table, along with the number, age of participants and specific diagnosis. The values of OPN we think will increase the bulk of the table making less readable, since we have added the HR and OR and the age. Moreover, different research groups have used different scales of OPN, which will further increase the mess.
- The authors should include a separate section to discuss therapeutic options for inhibiting or activating osteopontin in atherosclerotic diseases. Are there any ongoing clinical trials aimed at inhibiting or activating osteopontin in this patient group?
We really appreciate the reviewer’s suggestion and we have meticulously searched for clinical trials investigating the impact of net OPN inhibitors or activators. To our knowledge, there is no study aiming at pharmaceutical inhibition or activation of OPN in ASCVDs. We have added a related statement and references (lines 240-246).
- The authors should create a diagram illustrating the osteopontin signaling pathways in the pathogenesis of atherosclerotic diseases.
We have created a diagram illustrating the interaction of OPN with atherosclerosis pathways.
Round 2
Reviewer 1 Report
Comments and Suggestions for Authors
Concerns have been properly addressed.
Reviewer 2 Report
Comments and Suggestions for Authors
Dear Authors, thank you for your reply to my comments.